# Stochastic simulations of self-organized elastogenesis in the developing lung

Xiru Fan[1,2], Cristian Valenzuela[3], Weijing Zhao[4], Zi Chen[3]*, Dong Wang[1,2]*, Steven J. Mentzer[3]*

**1** State Key Laboratory of Mechanical System and Vibration, School of Mechanical Engineering, Shanghai Jiao Tong University, Shanghai, China, **2** Meta Robotics Institute, Shanghai Jiao Tong University, Shanghai, China, **3** Laboratory of Adaptive and Regenerative Biology, Brigham & Women's Hospital, Harvard Medical School, Boston, Massachusetts, United States of America, **4** Department of Endocrinology and Metabolism, Shanghai Sixth People's Hospital Affiliated to Shanghai Jiao Tong University School of Medicine, Shanghai, China

☯ These authors contributed equally to this work.

\* zchen33@bwh.harvard.edu (ZC); wang_dong@sjtu.edu.cn (DW); smentzer@bwh.harvard.edu (SJM)

## Abstract

In the normal lung, the dominant cable is an elastic "line element" composed of elastin fibers bound to a protein scaffold. The cable line element maintains alveolar geometry by balancing surface forces within the alveolus and changes in lung volume with exercise. Recent work in the postnatal rat lung has suggested that the process of cable development is self-organized in the extracellular matrix. Early in postnatal development, a blanket of tropoelastin (TE) spheres appear in the primitive lung. Within 7 to 10 days, the TE spheres are incorporated into a distributed protein scaffold creating the mature cable line element. To study the process of extracellular assembly, we used cellular automata (CA) simulations. CA simulations demonstrated that the intermediate step of tropoelastin self-aggregation into TE spheres enhanced the efficiency of cable formation more than 5-fold. Similarly, the rate of tropoelastin production had a direct impact on the efficiency of scaffold binding. The binding affinity of the tropoelastin to the protein scaffold, potentially reflecting heritable traits, also had a significant impact on cable development. In contrast, the spatial distribution of TE monomer production, increased Brownian motion and variations in scaffold geometry did not significantly impact simulations of cable development. We conclude that CA simulations are useful in exploring the impact of concentration, geometry, and movement on the fundamental process of elastogenesis.

## Author summary

The dominant cable in a healthy lung is made up of elastin fibers bound to a protein scaffold and helps maintain alveolar shape and lung volume during exercise. Recent research on postnatal rat lungs indicates that elastogenesis, the development of this cable, is a self-organized process within the extracellular matrix. Elastogenesis is difficult to study by conventional approaches but can be usefully explored by cellular automata simulations. First, elastogenesis involves discrete interactive variables—such as TE monomers and TE

**Data Availability Statement:** The software code of the model is available at https://github.com/Wang-s-Lab/CA_for_Self-organized_Elastogenesis.

**Funding:** This work is supported by National Key Research and Development Program of China Grant No. 2022YFB4700900 (DW); NIH Grant

HL134229, HL007734, and CA009535 (SJM); National Natural Science Foundation of China Grant No. 52275025 (DW); and Interdisciplinary Program of Shanghai Jiao Tong University Grant No. YG2021QN105 (WJZ). The funders had no role in study design, data collection and analysis, decision to publish, or preparation of the manuscript.

**Competing interests:** The authors declare that they have no competing interests.

spheres—that bind to a protein scaffold. Conceptualizing large mutually interactive populations, particularly difficult in the compressed timeframe of the cable development, is enhanced with simulations. Second, the length scale and geometry of our simulations is a reasonable reflection of the primary septa and the protein scaffold. This scale consistency suggests that the short-range planar interactions in our simulations will faithfully contribute to the large-scale 3-dimensional patterns observed in the developing lung. Finally, elastogenesis appears to be a self-organized process independent of supervised interactions. The simple rules explored in these simulations may not only reflect normal development but also suggest perturbations potentially associated with congenital lung diseases.

## 1. Introduction

Elastin fibers are ubiquitous in the extracellular matrix (ECM) of vertebrate tissues. Elastin fibers are an important structural component of the skin, lungs, tendons, cartilage, and cardiovascular system [1]. The elasticity, stability, and durability of elastin fibers are essential for complex functions such as lung ventilation [2]. In the normal lung, the dominant cable is an elastic "line element" composed of elastin fibers bound to a protein scaffold [3]. The cable line element originates in the hilar airways and ends in the subpleural lung [3] (Fig 1). Weibel has described the line element as an "ingenious" fiber continuum that supports the conducting airways as well as the fragile septa of the alveolar walls [4–6]. In children, defects in the cable line element can lead to the structural dysfunction associated with bronchopulmonary dysplasia [7]. In adults, disruption of the cable leads to the anatomic and functional limitations associated with emphysema [8].

The cable line element plays a central role in the alveolar phase of lung development. Alveolarization is the process that forms secondary alveolar septa by the lifting new tissue ridges from primitive primary septa. In rats, a species in which alveolar septation occurs postnatally [9], the cable line element forms between postnatal day 4 (P4) and day 14 (P14). Valenzuela and colleagues demonstrated that the process is characterized by a blanket of tropoelastin (TE) spheres detectable in the primary septa on postnatal day 4 (P4) [10] (Fig 2). The ubiquitous spheres had a mean diameter of 2 um and were uniformly distributed in primary alveolar septa [10]. By P14, the tropoelastin appeared to be incorporated into the mature cable line element in a process termed elastogenesis.

An intriguing observation was the spatial relationship of tropoelastin spheres and the protein scaffolding within the primary septa of rat pups [10]. Between P4 and P14, light and electron microscopy demonstrated tropoelastin spheres and elastin monomers linked to the protein scaffold. In contrast, Valenzuela and colleagues found no consistent relationship between the tropoelastin spheres and parenchymal cells. Reminiscent of the observations of elastin self-assembly in vitro [11,12], these findings indicated that the cable was self-organized by physical and chemical processes within the extracellular matrix.

To explore the fundamental process of elastogenesis, we employed cellular automata (CA) simulations. The simulations explored the concentration, geometry and movement of topoelastin spheres in the extracellular assembly of the cable line element.

## 2. Results

### 2.1. Elastin aggregation and protein binding

In the early postnatal period (P1-P4), TE monomers in the extracellular space can bind to the protein scaffold either before or after self-aggregation into TE spheres [10]. In our simulation

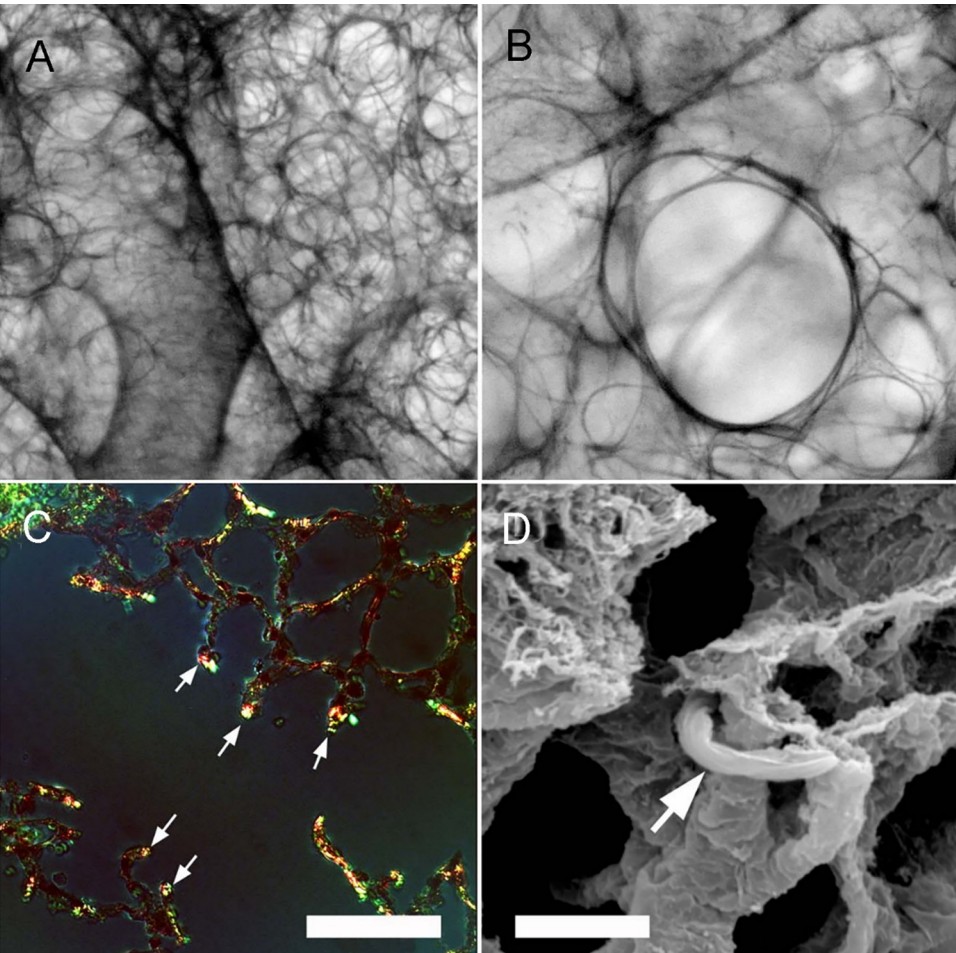

**Fig 1.** Lung elastic line element examined in a decellularized lung by light (A,B), fluorescence (C) and scanning electron microscopy (SEM) (D). (C) Fluorescence microscopy of thin tissue sections stained with Sirius red demonstrating red/orange collagen birefringence as well as green elastin staining at septal tips (white arrows) (bar = 100 um). (D) SEM of the line element demonstrated a central cable encased in basement membrane (bar = 2 um). Images from Wagner et al are reproduced with permission of Wiley & Sons [3].

(S1 Movie), a plateau was observed reflecting a dynamic balance between TE monomer generation rate and TE binding to the protein scaffold (both before and after self-aggregation) (Fig 3A). After the plateau, there was an expected increase in scaffold-associated TE and a commensurate decline in free TE monomer concentration. The average size of the TE spheres decreased as the available TE monomers were bound. The efficiency of TE binding to the protein scaffold suggested an 800-time step process interval; that is, a time interval compatible with the experimental observations of 8–10 days for the development of the mature cable [10]. Importantly, if TE monomers did not aggregate into TE spheres—and the process relied upon TE monomer binding alone—the formation of the cable line element required more than 45 days. This observation supported the role of TE spheres as critical intermediates in the formation of the cable line element.

## 2.2. Spatial distribution of TE monomer production

To assess the impact of the spatial distribution of TE monomer production, our simulation varied the spatial distribution of TE monomer production over a 20-fold range (Fig 4 and S2 Movie).

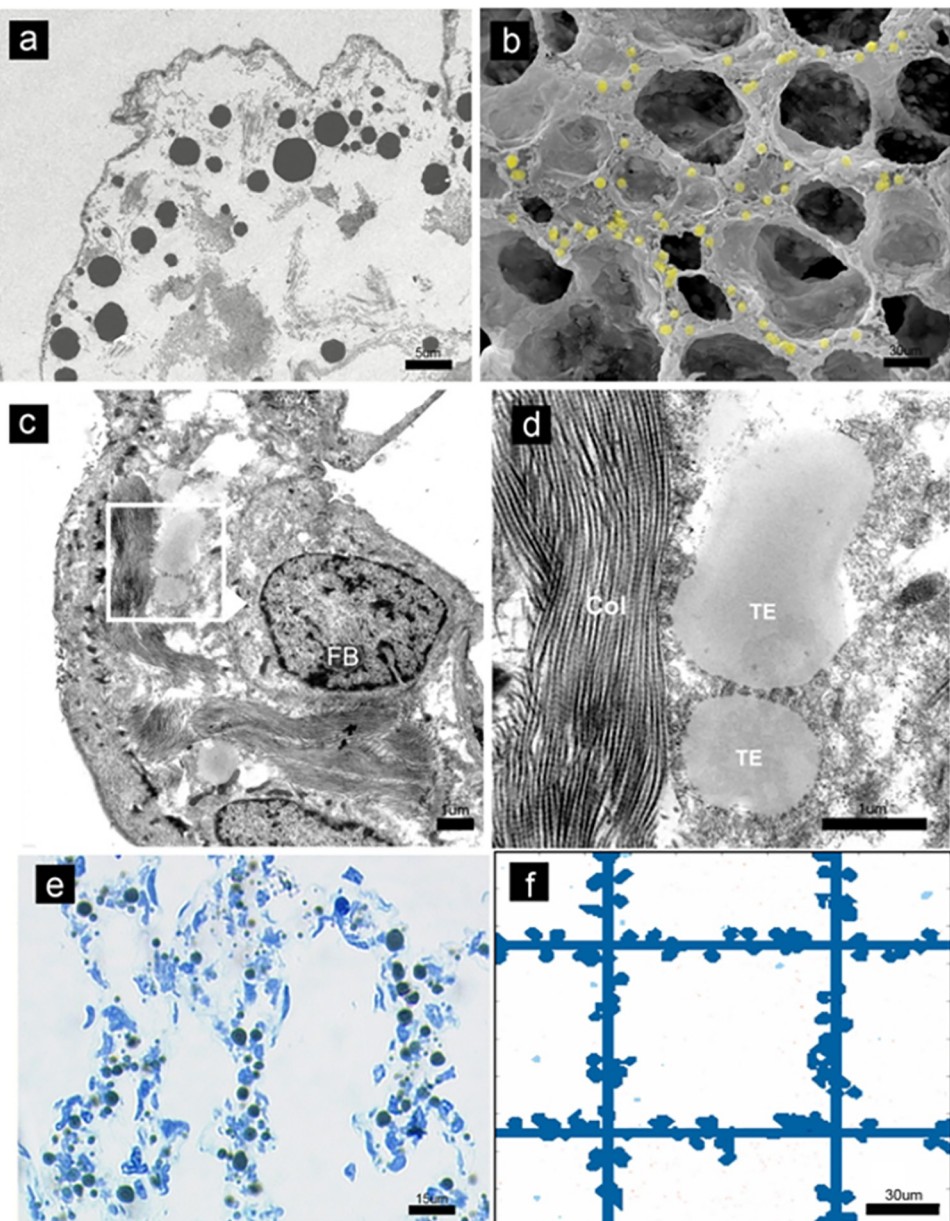

**Fig 2. Length scales in vivo and in silico.** (a) Numerous electron-dense TE-spheres are seen within the primary septa of the rat lung by transmission electron microscopy (TEM). (b) Scanning electron microscopy demonstrated that the TE-spheres were located within the primary sept (P4, shown; yellow pseudocoloring for presentation purposes). (c) TEM demonstrating prominent extracellular fiber interaction with the TE-spheres in the lung (FB = fibroblast). (d) Magnified view of TE-spheres interacting with the protein scaffold (Col = collagen; TE = tropoelastin). (e) Azure blue staining demonstrating the distribution of the spheres within the primary septa. (f) The length scale of the simulations was designed to reflect the experimental observations. The images reflect most steps in the simulation: step 3 (panel b and c); step 4 (panel d and e). Images from Valenzuela et al are reproduced with permission of Wiley & Sons. [10].

The TE monomer production rate was varied so that the total number of TE monomers was kept constant. The more concentrated production of TE monomers demonstrated a minimal peak preceding a plateau (Fig 4A, black); however, there was little difference between spatial production conditions (Fig 4B). The spatial distribution of TE monomer production—analogous to the

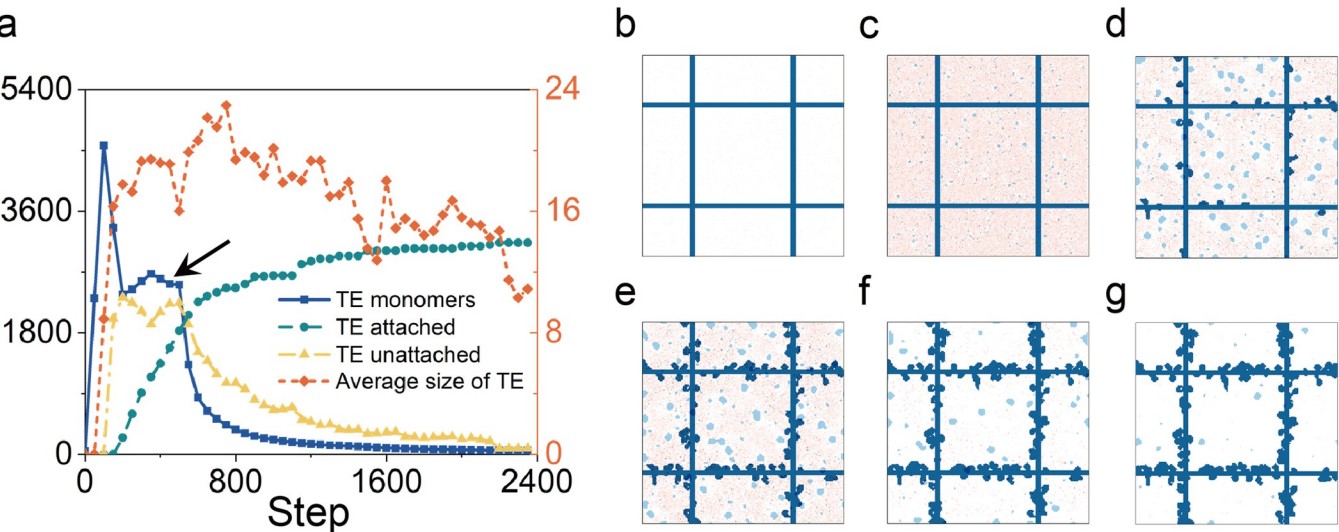

**Fig 3. Overview of TE dynamics with standard parameters.** (a) The production of TE monomers (dark blue) was associated with a dynamic plateau (arrow). The aggregation of TE monomers or TE monomer binding to the protein scaffold resulted in a subsequent plateau of bound elastin (green) and reciprocal decline in unbound TE monomers (yellow). The average size of TE aggregated decreased with time (orange). The simulation images at (b) step = 1, (c) step = 100, (d) step = 200, (e) step = 500, (f) step = 800, and (g) step = 2400. The red color represents the contents of the TE monomer. Each light blue sphere represents a TE. Dark blue lines represent the collagen fibers. Dark blue circle presents the attached TE spheres.

spatial distribution of cells secreting TE monomers—appeared to have only a modest impact on the efficiency of elastin cable formation.

### 2.3. Temporal distribution of TE monomer production

Experimental data indicated that the TE spheres appeared between P1 and P4 [10]. To assess the impact of the rate of TE monomer production, we simulated TE monomer production over a 4-fold range of uniform rates (S3 Movie). Production rates were assessed for their impact on the total area of TE monomer and TE spheres unattached and attached to the protein scaffold (Fig 5). The highest production rate (shorter production interval) demonstrated an early aggregate plateau and rapid decline (Fig 5A). In contrast, lower production rates (longer production interval) had a prolonged plateau. Notably, the higher production rate had a supra-additive effect on scaffold binding suggesting the beneficial impact of accelerated production rates (Fig 5C). Qualitatively similar results were obtained with a variable (Gaussian) rate of TE monomer production (S1 and S2 Figs and S4 Movie). The efficiencies gained from higher production rates suggests an adaptive advantage for the burst of elastin transcriptional activity observed in the postnatal period [13].

### 2.4. Brownian motion

The potential impact of self-diffusion, the random motions of TE monomers, was assessed over a 2.5-fold range (S5 Movie). The impact of Brownian motion on the efficiency of TE monomer aggregation (Fig 6A) and scaffold binding (Fig 6B) was limited. The modest impact was consistent irrespective of the rate of TE monomer production.

### 2.5. TE aggregation affinity

TE monomer aggregation and scaffold binding affinities reflect the physicochemical milieu of elastogenesis. In our simulation, this aggregation and binding affinity was reflected by

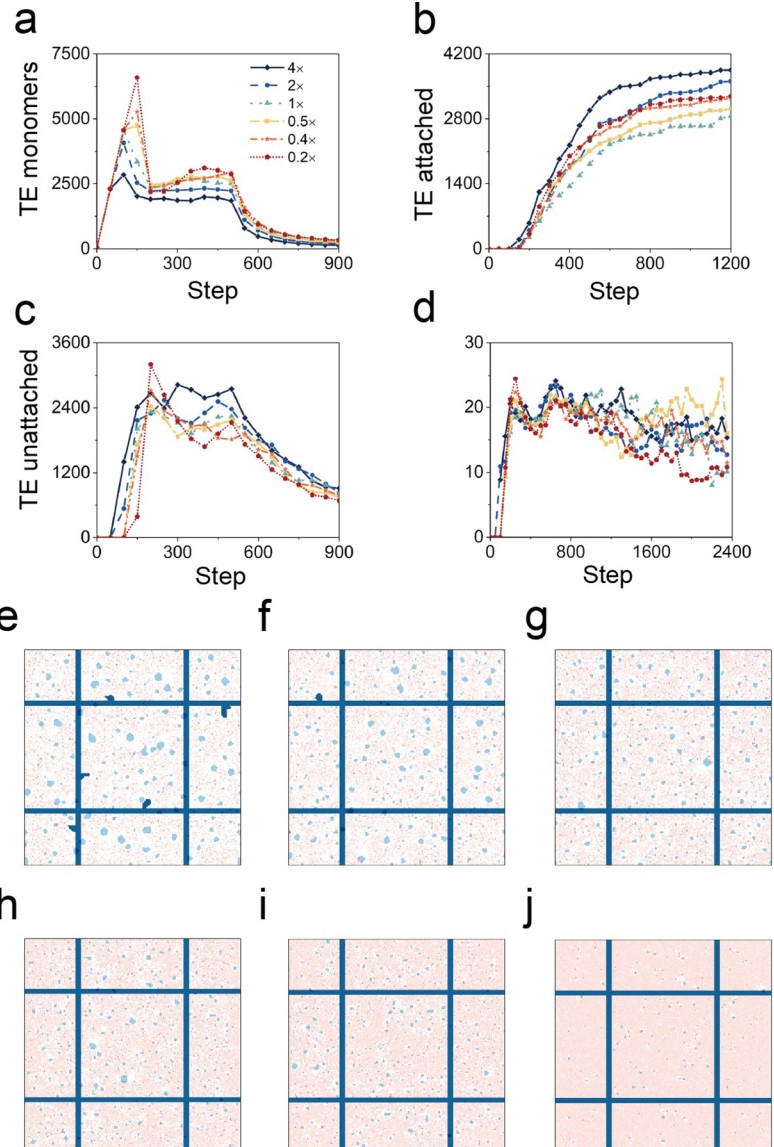

**Fig 4. Effects of the spatial distributions of TE monomer production.** The total number of TE monomers being produced was kept constant. The spatial concentration of TE monomer production was varied over a 20-fold range. (a) The total area of TE monomers was plotted as a function of spatial distribution with the less concentrated conditions (red) demonstrating a higher peak of TE monomer area than the more concentrated conditions (blue). The number of (b) attached TE spheres, (c) unattached TE spheres, and (d) the average size of unattached TE spheres varied only modestly with spatial distribution. Simulation images at step = 150 with different combinations of spatial distribution and production rates; that is, 4x (e), 2x (f), 1x, (g) 0.5x, (h) 0.4x (i), and 0.2x (j).

variation in the critical density of particle aggregation and scaffold binding. A low critical density for aggregation and binding was analogous to high binding affinity. Our simulations were analyzed over a 2-fold range of critical density (S3 and S4 Figs, and S6 Movie). A low critical density (high binding affinity) was associated with a limited number of TE monomers—reflecting the rapid increase in TE spheres (Fig 7C). In contrast, high critical density was associated with a high concentration of TE monomers. These results suggest that impaired binding affinity (high critical density) would likely be clinically associated with patchy or incomplete elastogenesis.

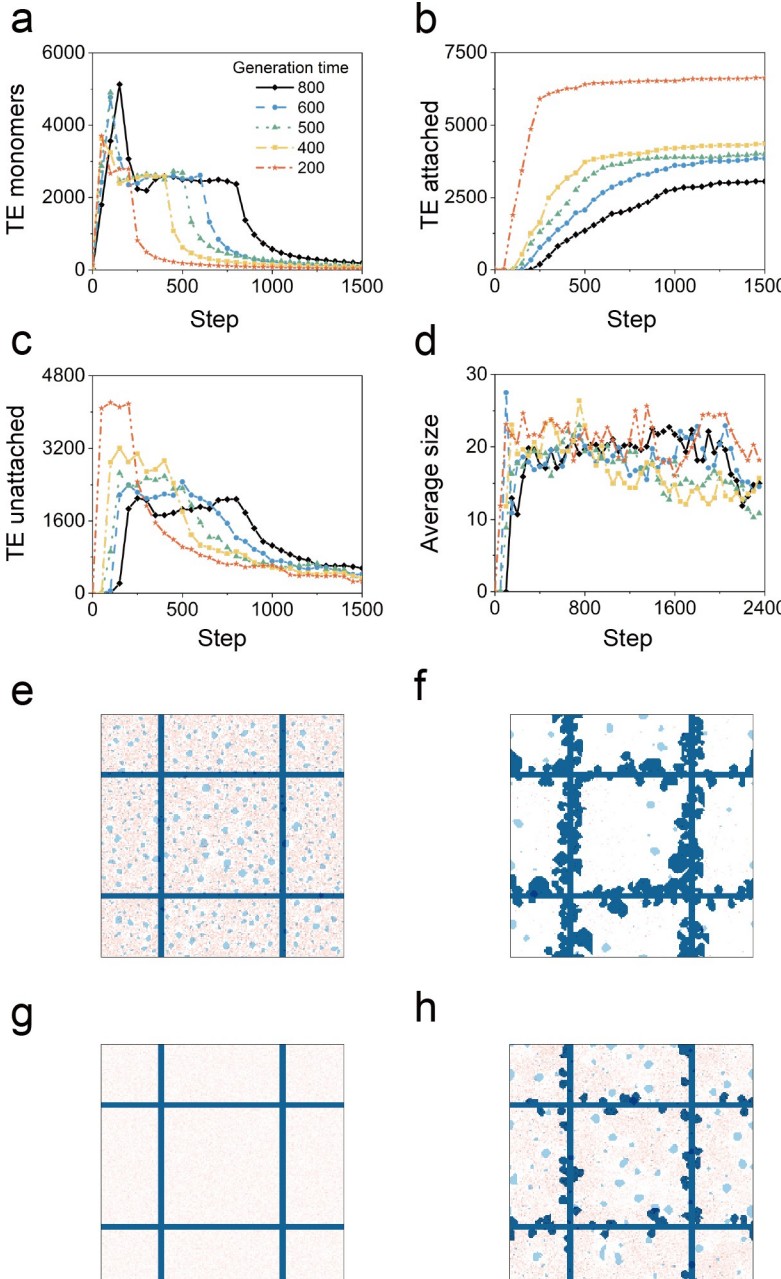

**Fig 5. Effects of varied production rates of the TE monomers.** The TE monomers were produced with uniform distribution over a 4-fold time scale. Simulation images at (e) step = 50, (f) step = 750 when the production time is 200 steps and at (g) step = 50, (h) step = 750 when the production time is 800 steps.

## 2.6. Scaffold geometry

The primary septa provide an anatomic boundary for the development of the cable line element. To explore the potential importance of protein scaffold geometry, we tested a variety of scaffold shapes (S5 Fig and S7 Movie). The geometric shapes had no effect on the efficiency nor the appearance of the final scaffold. These results suggest that our findings are not limited by our idealized scaffold geometry.

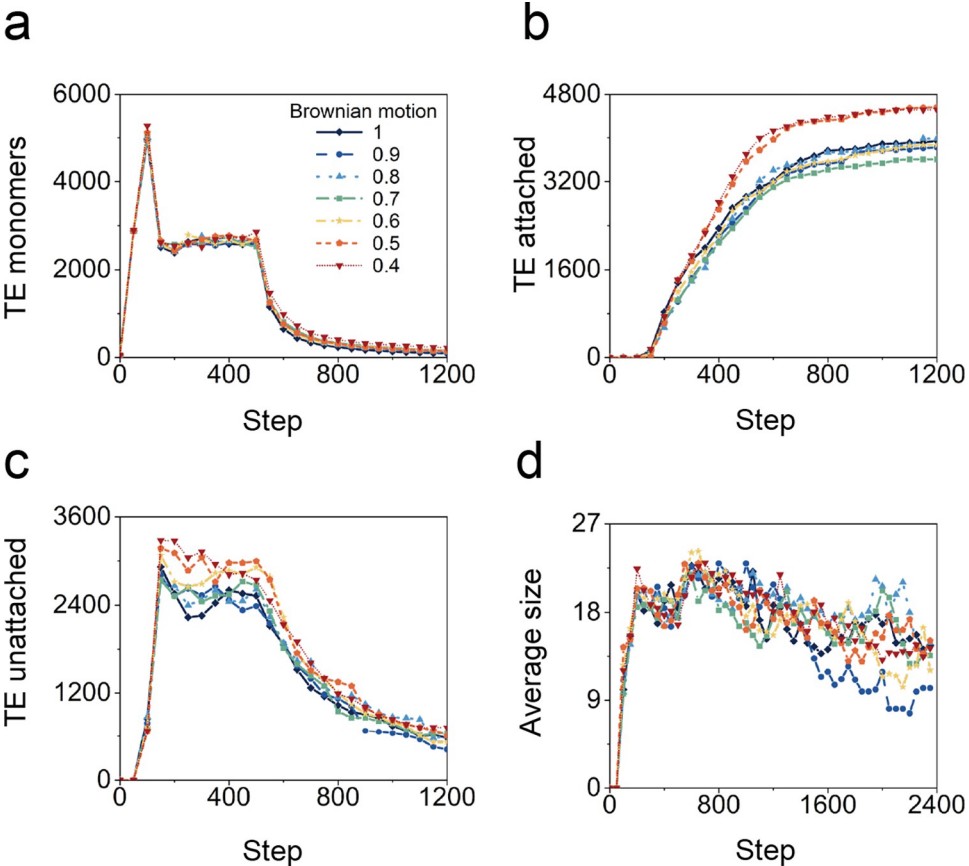

**Fig 6. Effects of the Brownian motion when the spatial distribution of TE particles production is uniform.** The dependence of the number of (a) TE particle, (b) attached TE spheres, (c) unattached TE spheres, and (d) the average size of unattached TE spheres on time step with different production time.

## 3. Discussion and conclusion

Previous work has shown that cable line element is assembled in the extracellular space—apparently independent of cellular instruction [10]. Here, we use cellular automata simulations to show that the extracellular assembly of the cable line element is consistent with a self-organized process dependent upon the spatial and temporal distribution of extracellular TE spheres.

Elastogenesis is a fundamental developmental process that is difficult to study by conventional approaches but can be usefully explored by cellular automata simulations. First, elastogenesis involves discrete interactive variables—such as TE monomers and TE spheres—that bind to a protein scaffold. Conceptualizing large mutually interactive populations, particularly difficult in the compressed timeframe of the cable development, is enhanced with simulations. Second, the length scale and geometry of our simulations is a reasonable reflection of the primary septa and the protein scaffold. This scale consistency suggests that the short-range planar interactions in our simulations will faithfully contribute to the large-scale 3-dimensional patterns observed in the developing lung. Finally, elastogenesis appears to be a self-organized process independent of supervised interactions [10]. The simple rules explored in these simulations may not only reflect normal development [14,15] but also suggest perturbations potentially associated with congenital lung diseases [16].

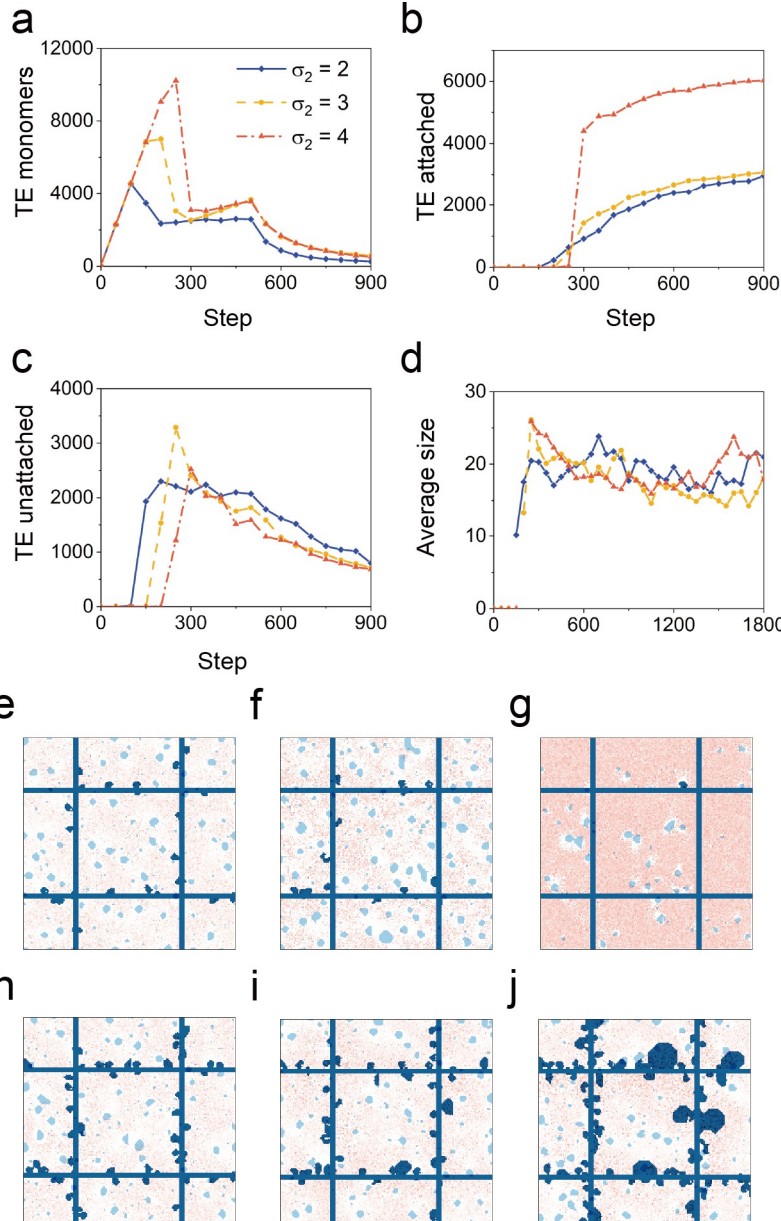

**Fig 7. Effects of aggregation affinity ($\sigma_2$).** The dependence of the number of (a) TE particle, (b) attached TE spheres, (c) unattached TE spheres, and (d) the average size of unattached TE spheres on time step with different production time. Simulation images at (e) step = 250, (h) step = 400 when $\sigma_2 = 2$ at (f) step = 250, (i) step = 400 when $\sigma_2 = 3$, and at (g) step = 250, (j) step = 400 when $\sigma_2 = 4$.

An intriguing observation from these simulations was the impact of the TE spheres on the efficiency of cable development. Secreted from elastogenic cells as a 60–70 kDa protein, TE is the soluble precursor of elastin. The TE monomer has alternating hydrophobic and hydrophilic domains [17]. The interactions between hydrophilic and hydrophobic domains—a process called coacervation—facilitates the self-assembly of TE into quantized spheres [18]. The TE spheres have been postulated to function as intermediates in elastin macroassembly [19]. Lysine residues on the TE sphere surface facilitate the lysyl oxidase-dependent cross-linking of TE spheres to not only to other spheres, but also extracellular scaffold proteins [20]. Our

simulations suggested an adaptive advantage of the process of coacervation and the development TE sphere intermediates; namely, the formation of TE spheres enhanced the efficiency of elastin binding to the protein scaffold by an estimated 5-fold.

A limitation of this work is our simplification of the experimental process of structural self-organization. The initial steps (1 and 2) in the process—the generation of TE monomers and aggregation of TE monomers—are below the limits of our microscopic resolution. Nonetheless, we know from molecular data that TE monomer transcription is active during this phase [21]. A second limitation is our simplified assembly of the cable line element. The protein scaffold involves numerous extracellular components. A growing list of more than 30 different molecules, identified by microscopy and immunohistochemical approaches, have been found to interact with elastic fibers and microfibrils [22]. The interactions of these extracellular components is an important frontier in understanding the process of self-assembly.

The random and uniform distribution of the TE spheres at the onset of secondary septation suggests that it is the scaffold proteins, not the tropoelastin spheres, that are responsive to mechanical force distribution [10]. Tomoda et al. have shown that the collagen fiber orientation closely correlates with respiratory movement [23]. Pins and colleagues as well as Chow and coworkers have shown that the axial alignment of collagen fibers can be achieved by mechanical loading of the uncrosslinked fibers [24–26]. Fibrillin microfibrils also undergo gross changes in molecular configuration when subjected to stretch [27]. Although less well-studied than collagen, fibrillin microfibrils subjected to stretch undergo significant changes in molecular conformation [27]. The conformational effects of mechanical loading provides a useful explanation for the organ-sized length scales of the cable line element; that is, extracellular loads contribute to the efficiency of a structure that extends from the central airways to the subpleural matrix [3].

The results of our simulations—based on in vivo observations [10]—suggests that the cable line element self-organizes in the extracellular matrix. Although elastogenic cells such as fibroblasts, muscle cells and chondrocytes, produce tropoelastin, the physicochemical properties of the extracellular matrix contribute to tropoelastin formation and scaffolding interactions. The observations are reminiscent of in vitro elastin self-assembly [11,28]. We speculate that lung ventilation provides a motive force for this process. Finally, self-organized elastogenesis has practical implications for normal lung development and the pathologic processes of bronchopulmonary dysplasia. We anticipate this will be a promising area of future research.

## 4. Methods

### 4.1. Cellular automata simulation procedures

According to the experimental observations (Figs 1 and 2), the process of the structural self-assembly of collagen and elastin in the developing mammalian lung can be divided into the following steps: generation of the TE monomer; TE monomer aggregating into TE spheres; aggregation of TE spheres; Attachment of the TE spheres to the protein scaffold. We use cellular automata to simulate these steps. The code is implemented using MATLAB (2021a). The detailed simulation procedures are explained below.

### 4.2. Generation of tropoelastin monomer

TE spheres are formed by the self-assembly of TE monomers. The TE monomers are generated in a limited time (denoted by *generate_Time*) in the simulation. As the size of TE monomers is around 15 nm, we used a value of each square unit in the simulation domain to represent the contents of the tropoelastin monomer. We use red color to represent the TE monomers and use transparency to represent the

value magnitude. A lower transparency represents a higher value. In each step within *generate_Time*, the value of each unit increases by *generate_Tol* with the probability of *generate_rate*. Two types of generation values were used: a constant generation value or a Gaussian distributed generation value. The constant or Gaussian "generation values" distributions are over time. The total generated tropoelastin monomers are the same in the simulations with Gaussian distributions and with constant generate value.

As the TE monomers and TE undergo random Brownian motions, Brownian motions are included in the simulation. The Margolus neighborhood was used to mimic the Brownian motion of the TE monomer. The particle movement strategy is shown in S6 Fig. The domain was divided into blocks of size 2×2in each step. The blocks are aligned with the even grid at the even steps, while aligned with the odd grid for odd steps (S6a Fig). In each step, the values in the four units in a Margolus neighborhood rotate in clockwise or counterclockwise directions with equal possibilities (S6b Fig).

## 4.3. TE monomoer self-assembly into TE spheres

The tropoelastin monomer will undergo self-assembly to form tropoelastin. To model this process, the tropoelastin monomers are transited to tropoelastin spheres under two situations: (1) the value representing the monomer's content reaches a threshold $\sigma_1$, or (2) the values of the Moore neighborhood (composed of a central cell and the eight cells that surround it) added together to reach a threshold $\sigma_2$. Both situations are based on the reasonable hypothesis that tropoelastin spheres are formed when the content of the monomers exceeds a nucleation threshold [29]. In the simulation, each light blue sphere represents a TE. As the minimum diameter of the tropoelastin sphere is around 1 μm, the length of one unit is set as 1 μm. Brownian motion is also applied to the tropoelastin spheres.

## 4.4. Aggregation of tropoelastin

From the experimental observation that the diameters of tropoelastin spheres span from 1–6 μm (Fig 2), we can conclude that the tropoelastin spheres will aggregate to form larger coalesced tropoelastin spheres. In the simulation, if two tropoelastin spheres contact, they will adhere and form larger tropoelastin spheres. A division principle is used to prevent the overgrowth of the tropoelastin. If the size of a tropoelastin sphere is larger than $\delta_1$, it will split into two spheres, similar to the division of large water droplets. In the simulation, we choose $\delta_1 =$ ~$6.5^2 = 42$, corresponding to a full-sized coacervated TE with the diameter of around 6–7 μm, which agree well with the experimental observations that the elastin of the postnatal rats was detected as 1–6 μm tropoelastin spheres. $\delta_2$ represents the minimum size of the TE that can crosslink with the collagen scaffolds. We set $\delta_2 =$ ~$0.9\delta_1 = 38$, as most crosslinked TE are large from experimental observation.

## 4.5. Attachment of TE to the protein scaffold

The in vivo experiments demonstrate apparent crosslinking of the TE spheres to the protein scaffold (Fig 2). To simulate the adhering process, the following rule is used. If a coalesced tropoelastin is adjacent to the fibrillar structures and its size is larger than $\delta_2$, it will crosslink to the collagen scaffolds. The pseudocode for the cellular automata simulation are shown in Table 1. The standard parameters used in the simulation are shown in Table 2. Three different cable line shapes are used in the simulation. The cable lines are assumed to be straight lines which forms square, hexagon and triangular shapes, respectively. The linewidth is 4 pixels. The cable lines form square, hexagon or triangular periodic units with the same areas. Periodic boundary conditions are used.

**Table 1. Pseudocode of the cellular automata simulation.**

```
for 0 < t < total_Time do:
    if t < generate_Time do:
        generate particles with generate_Tol in constant rate generate_Rate
    end if
    for all particles in computational domain N×N do:
        particle move one step randomly using Margolus neighborhood
        if particle > σ₁ or sum(particles) > σ₂
            particle merge into a sphere
        end if
    end for
    for all Spheres in computational domain do:
        if size(sphereA)> δ₁
            sphereA split into two parts
        end if
        if size(sphere)> δ₂ and sphere is adjacent to the cable line do:
            sphere sticks to the cable
        else
            sphere moves one step randomly
        end if
    end for
end for
```

## 4.6. Time and length scale

The time and length scale of the simulation is discussed. As mentioned before, the size of a TE sphere is around 1 μm and is represented by 1 pixel (a unit cell). Therefore, the length scale is $\lambda_l$ = 1 μm/pixel. The length scale in experiments and simulations is shown in Fig 2. the TEM images of the TE of rats on postnatal days 4 are shown in Fig 2. Azure blue staining provides sufficient contrast to demonstrate the broad distribution of the sphere. Fig 2F shows the simulation image of TE. The computation domain is chosen as 200×200 μm². It can be seen that the length scale of the simulation agrees with the experimental observations.

Next, the time scale is studied. For a random walk, the mean squared displacement of a particle is proportional to the time interval [30]:

$$\lambda_l^2 = 2D\lambda_t, \tag{1}$$

where $D$ is the diffusion coefficient, $\lambda_t$ is the time scale. The diffusion coefficient of a particle can be calculated using the Stokes-Einstein equation in Brownian motion

$$D = \frac{k_B T}{3\pi\eta d}, \tag{2}$$

**Table 2. The standard parameters used in the simulation.**

| Symbol | Value |
|---|---|
| total_Time | 2400 |
| generate_Time | 500 |
| n | 200 |
| generate_Tol | 0.04 |
| generate_Rate | 0.032 |
| $\sigma_1$ | 0.7 |
| $\sigma_2$ | 2 |
| $\delta_1$ | 42 |
| $\delta_2$ | 38 |

where $k_B$ is the Boltzmann's constant. $T$ is the absolute temperature, $\eta = 1.9$ $Pa \cdot s$ is the viscosity coefficient of the surrounding liquid [31]. $d = \lambda_1$ is the particle diameter. According to the above Stokes-Einstein equation, small particle size, low viscosity of the surrounding fluid and high temperature result in faster motion. $D$ is calculated as $2.412 \times 10^{-16}$ m²/s. The time scale can then be calculated using Eq (1) as $\lambda_1 \approx 2072$ seconds = 0.024 days. Experiments show that a blanket of tropoelastin (TE) spheres is detectable in the primary septa on postnatal day 4 (P4) (Fig 2), and the tropoelastin appeared to be incorporated into the mature cable line element by P14. Therefore, the generate_Time of the tropoelastin monomers is between days 4 and 14. We choose the generate_Time as 500 in the simulation, which corresponds to 12 days.

Brownian motion simulations of multiple particles are conducted, as shown in S7 Fig. The particle trajectories are shown in S7a Fig. The dependence of the displacement squared on time is shown in S7b Fig. The displacement squared is equal to the x coordinate squared plus the y coordinate squared. The theoretical and simulated average mean squared displacements are plotted against the time step.

## Supporting information

**S1 Fig. Gaussian distributions of the generate rate.** (a) The probability density function of Gaussian distribution for N(0, 1²). (b) The probability density function of Gaussian distribution for N(0, 1²), N(0, 3²) and N(-2, 2²).
(TIF)

**S2 Fig. Effects of the generation time to TE monomers with Gaussian distributions.** The dependence of the number of (a) TE monomers, (b) attached TE spheres, (c) unattached TE spheres, and (d) the average size of unattached TE spheres on time step with different generation time.
(TIF)

**S3 Fig. Numerical results of simulations with different $\sigma_1$.** (a) total area of TE monomers; (b) total area of TEs attached to the collagen; (c) total area of TEs unattached to the collagen; (d) average area of TEs unattached to the collagen (ignore TEs smaller than 3 pixels).
(TIF)

**S4 Fig. Numerical results of $(\delta_1, \delta_2)$.** (a) total area of TE monomers; (b) total area of TEs attached to the collagen; (c) total area of TEs unattached to the collagen; (d) average area of TEs unattached to the collagen (ignore TEs smaller than 3 pixels).
(TIF)

**S5 Fig. Effects of collagen shapes.** (a) TE monomers, (b) attached TE spheres, (c) unattached TE spheres, and (d) the average size of unattached TE spheres on time step with different collagen shapes. The simulation image at step (e) 250 and (h) 400 with a square collagen shape, at step (f) 250 and (i) 400 with a hexagon collagen shape and at step (g) 250 and (j) 400 with a triangle collagen shape.
(TIF)

**S6 Fig. Tropoelastion monomer movement strategy.** (a) The 2×2 blocks of the Margolus neighborhood; consecutive steps alternate between the even grid and the odd grid. (b) In each step, the entire block is rotated in clockwise or counterclockwise directions with equal possibilities.
(TIF)

**S7 Fig. Brownian motion simulation of multiple particles. (a) The** particle position trajectories. (b) The comparisons between the theoretical and average displacements squared versus time. The displacement squared of each particle is also shown.
(TIF)

**S1 Movie. Cellular automata simulation of TE dynamics with standard parameters. using standard parameters.**
(MP4)

**S2 Movie. Effects of the spatial distributions of TE monomer production.**
(MP4)

**S3 Movie. Effects of varied production rates of the TE monomers (uniform rate).**
(MP4)

**S4 Movie. Effects of varied production rates of the TE monomers (Gaussian rate).**
(MP4)

**S5 Movie. Effects of the Brownian motion when the spatial distribution of TE particles production is uniform.**
(MP4)

**S6 Movie. Effects of aggregation affinity ($\sigma_2$).**
(MP4)

**S7 Movie. Effects of collagen shapes.**
(MP4)

## Author Contributions

**Conceptualization:** Zi Chen, Dong Wang, Steven J. Mentzer.

**Data curation:** Cristian Valenzuela.

**Formal analysis:** Xiru Fan, Cristian Valenzuela, Weijing Zhao.

**Funding acquisition:** Weijing Zhao, Dong Wang, Steven J. Mentzer.

**Investigation:** Weijing Zhao.

**Methodology:** Xiru Fan, Cristian Valenzuela, Dong Wang.

**Supervision:** Dong Wang, Steven J. Mentzer.

**Visualization:** Xiru Fan.

**Writing – original draft:** Xiru Fan.

**Writing – review & editing:** Dong Wang, Steven J. Mentzer.

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
