## [Decision Letter · Decision Letter 0]

27 Apr 2023

Dear Prof. Wang,

Thank you very much for submitting your manuscript "Stochastic Simulations of Self-organized Elastogenesis in the Developing Lung" for consideration at PLOS Computational Biology.

As with all papers reviewed by the journal, your manuscript was reviewed by members of the editorial board and by several independent reviewers. In light of the reviews (below this email), we would like to invite the resubmission of a significantly-revised version that takes into account the reviewers' comments.

We cannot make any decision about publication until we have seen the revised manuscript and your response to the reviewers' comments. Your revised manuscript is also likely to be sent to reviewers for further evaluation.

Sincerely,

Philip K Maini

Academic Editor

PLOS Computational Biology

Jason Haugh

Section Editor

PLOS Computational Biology

Reviewer's Responses to Questions

**Comments to the Authors:**

Reviewer #1: This work addresses the very interesting topic of how elastin cables form, placed within the context of early alveolarization in the neonatal rat lung. The authors use some innovative computational approaches to model this process from the formation of early elastin spheres into elastin cables that are required to stabilize alveoli.

Overall I think its a very useful contribution.

I think it could perhaps be improved by thinking some more about what elastin is actually doing during alveologenesis.

Elastin becomes organized as a set of interlinked rings that surround and stabilize the mouths of millions of alveoli.

When the lung is considered in 3D as opposed to 2D it is actually rather obvious that this is what is really going on as opposed to the "erection of septae" by cables lying in the "septal tips" as put forth and perpetuated by Weibel, Schittny et al. These interlocking rings also account for the elastic interdependence of the lung alveoli as well as for the failure of formation of alveoli in BPD and or in Marfan's Ehlers-Danols and other elastin matrix diseases as well as for degeneration of alveoli in emphysema.

These ideas may be beyond the scope of the present paper, but I thought the authors may find them interesting to consider and perhaps incorporate into their thinking and modeling.

I have included a couple of papers FYI:

Conserved Mechanisms in the Formation of the Airways and Alveoli of the Lung.

Warburton D.

Front Cell Dev Biol. 2021 Jun 15;9:662059. doi: 10.3389/fcell.2021.662059. eCollection 2021.

PMID: 34211971

Spatial and temporal changes in extracellular elastin and laminin distribution during lung alveolar development.

Luo Y, Li N, Chen H, Fernandez GE, Warburton D, Moats R, Mecham RP, Krenitsky D, Pryhuber GS, Shi W.

Sci Rep. 2018 May 29;8(1):8334. doi: 10.1038/s41598-018-26673-1.

PMID: 29844468

Reviewer #2: This article presents a cellular automata model of the development of lung tissue. This is a geometrically simplified but illustrates that structures that may be observed in the lung could be established from cellular self-organisation, which is interesting for the general readership of this journal. However, there are some missing details in the methods making it difficult to interpret the model, and there is not a clear link regarding how this model would/could be tested against experimental data.

It would be useful to see a more detailed description of the line element/cable model for fibres, including reference to original studies proposing this model, or include a diagram of what is meant by a “dominant cable”. I don’t think it will be evident to all readers of this journal what is meant by a cable line element - that this is a model for the structure of the elastin fibres is not clear.

In the text Fig 1 is stated to show the process of structural self-organization experimentally: 1) generation of TE monomer, 2) aggregation into spheres, 3) aggregation of spheres, 4) attachment to scaffold. I cannot see this process in Fig 1. What I see in the figure is the spatial scales of importance to the process, but not the process itself. For example ,step 1 and 2 is not in the figure, the smallest scale is the spheres themselves. It is not clear which panels show step 3, I think (b/c), and then (d/e) show step 4, but this is not indicated in the caption. F. shows the model itself, but at this stage in the methods its not clear what we are seeing here, what are the lines?

I think the description of the scaffold in your model is missing from the paper. You do state that your simulation is happening in a square, but I had to assume in the figures that the lines represent some sort of scaffold material. The geometry imposed on your model should be described early in the methods. Geometry is mentioned in the results but nowhere in the methods. That geometry is not clearly stated in the methods makes it difficult to then interpret the steps in the cellular automata.

4.2 tropoelastin monomer: Are your constant or gaussian “generation values” distributions over time or space? How do you define the gaussian distributions (same mean as constant value? Based on some data?)?

Where you discuss experimental data in the methods there should be a citation to the source of that data, are they all ref [10]?

Section 4.4 “the diameters of tropoelastic spheres range from 1-6um”, section 4.6 “the size of a TE sphere is about 1 mm”. These length scales are quite different. I think this might be a typo.

I can see that the model is derived based on experimental data, but not so clearly the relationship back to the data on how the model could be validated or how it could guide experimental work in the future. This is hinted at in the fourth paragraph of the discussion but more explicit statements would be beneficial.

Pseudo-code is provided, I can’t see any information on, or link to, how this code was implemented, i.e. what software or tools were used?

Reviewer #3: The authors investigated the process of extracellular assembly using cellular automata (CA) simulations. The effect of the rate of the tropoelastin production on the efficiency of scaffold binding was presented. The results demonstrated that the intermediate step of tropoelastin self-aggregation into tropoelastin (TE) spheres increased the efficiency of cable formation by a fact of more than five. The rate of tropoelastin synthesis and its affinity for the protein scaffold also significantly influenced cable development, providing insight into the biophysical mechanism underlying the formation of elastin fibers in living things. The results are interesting. Following areas needs to be improved:

Criteria behind choosing the specific values of δ_1=42 for the division principle, and δ_2 for the crosslinking of tropoelastin and collagen scaffolds is not clear. Please elaborate.

The selection of standard time parameters used in the simulation in Table 2 is not clear. Please discuss.

Please provide further discussion on potential limitations or assumptions that were made in the simulations. While the results appear promising, it would be valuable to have a better understanding of the scope and limitations of the simulations to better contextualize the findings.

Please explain the motivation behind selecting specifically three different collagen shapes, square, hexagon, and triangle. Did you change any parameters other than geometry for their simulations? Please explain how you quantified that there is no effect on the efficiency of the final scaffold by changing the geometry.

In Section 4.3, you mentioned two situations for the transition of tropoelastin monomers to spheres. “Both situations are based on the reasonable hypothesis that tropoelastin spheres are formed when the content of the monomers is high enough.” How high? Please be specific.

Figure S7, sub-figures (h) (i) (j) are not detailed in the caption.

Please fix the typos; e.g., “tropoleastin”

**Have the authors made all data and (if applicable) computational code underlying the findings in their manuscript fully available?**

Reviewer #1: Yes

Reviewer #2: **No: **Pseudo code offered. No detail provided on model implementation.

Reviewer #3: None

PLOS authors have the option to publish the peer review history of their article (what does this mean?). If published, this will include your full peer review and any attached files.

Reviewer #1: No

Reviewer #2: **Yes: **Alys Clark

Reviewer #3: No
---

## [Decision Letter · Decision Letter 1]

26 May 2023

Dear Prof. Wang,

We are pleased to inform you that your manuscript 'Stochastic Simulations of Self-organized Elastogenesis in the Developing Lung' has been provisionally accepted for publication in PLOS Computational Biology.

Best regards,

Philip K Maini

Academic Editor

PLOS Computational Biology

Jason Haugh

Section Editor

PLOS Computational Biology

Reviewer's Responses to Questions

**Comments to the Authors:**

Reviewer #1: You have done a decent job of revising the paper.

I still think your conclusions are a bit weak but that’s a matter of taste I suppose.

Reviewer #2: Thank you for taking the time to address my comments, the revised manuscript addresses my concerns.

Reviewer #3: The authors have adequately addressed my comments. I don't have additional comments.

**Have the authors made all data and (if applicable) computational code underlying the findings in their manuscript fully available?**

Reviewer #1: Yes

Reviewer #2: Yes

Reviewer #3: None

PLOS authors have the option to publish the peer review history of their article (what does this mean?). If published, this will include your full peer review and any attached files.

Reviewer #1: **Yes: **David Warburton

Reviewer #2: **Yes: **Alys Clark

Reviewer #3: No

---

## [Editor Report · Acceptance letter]

9 Jun 2023

PCOMPBIOL-D-23-00244R1 

Stochastic Simulations of Self-organized Elastogenesis in the Developing Lung

Dear Dr Wang,

I am pleased to inform you that your manuscript has been formally accepted for publication in PLOS Computational Biology. Your manuscript is now with our production department and you will be notified of the publication date in due course.

With kind regards,

Lilla Horvath
